

# The Seasonal Variability in the Semidiurnal Internal Tide; A Comparison between Sea Surface Height and Energetics

Harpreet Kaur[1], Maarten C. Buijsman[1], Zhongxiang Zhao[2], and Jay F. Shriver[3]

[1]School of Ocean Science and Engineering, University of Southern Mississippi, Stennis Space Center, Mississippi, USA
[2]Applied Physics Laboratory and School of Oceanography, University of Washington, Seattle, Washington, USA
[3]Ocean Dynamics and Prediction Branch, Naval Research Laboratory, Stennis Space Center, Mississippi, USA

**Correspondence:** Harpreet Kaur (harpreet.kaur@usm.edu)

**Abstract.** We investigate the seasonal variability of the semidiurnal internal tide steric sea surface height (SSSH) and energetics using 8-km global Hybrid Coordinate Ocean Model (HYCOM) simulations with realistic forcing and satellite altimeter data. In numerous previous studies, SSSH has been used to explore the seasonal changes in internal tides. For the first time, we compare the seasonal variability of the semidiurnal internal tide SSSH with the seasonal variability of the semidiurnal baroclinic energetics. We explore the seasonal trends in SSSH variance, barotropic to baroclinic conversion rate, kinetic energy, available potential energy, and pressure flux for the semidiurnal internal tides. We find that the seasonal cycle of monthly semidiurnal SSSH variance in the Northern Hemisphere is out of phase with the Southern Hemisphere. This north-south phase difference and its timing are in agreement with altimetry. The amplitudes of the seasonal variability in SSSH variance are about 10-15% of their annual-mean values when zonally averaged. The normalized amplitude of the seasonal variability is higher for the SSSH variance than for the energetics. The largest seasonal variability is observed in Georges Bank and the Arabian Sea, where the seasonal trends of monthly SSSH variance and energetics are in phase. However, outside these hotspots, the seasonal variability in semidiurnal energetics is out of phase with semidiurnal SSSH variance and a clear phase difference between the Northern and Southern Hemispheres is lacking. While the seasonal variability in semidiurnal energy is driven by seasonal changes in barotropic to baroclinic conversion, semidiurnal SSSH variance is also modulated by seasonal changes in stratification. Surface intensified stratification at the end of summer enhances the surface perturbation pressures, which enhance the SSSH amplitudes.

## 1 Introduction

The interaction between surface tides and bathymetry, in the presence of density stratification, leads to the generation of internal tides (Huthnance, 1981; Baines, 1982; Gerkema and Zimmerman, 2008; Buijsman et al., 2020). These internal waves propagate away from their generation sites and eventually dissipate. It is important to study internal tides as their dissipation may cause water mass mixing (Waterhouse et al., 2014; Melet et al., 2016), affecting the global ocean circulation (Munk and Wunsch, 1998; St. Laurent and Garrett, 2002). The mixing caused by internal tides also affects sediment transport (Sinnett et al., 2018) and the dispersal of nutrients (Tuerena et al., 2019). Internal tides from the deep ocean propagate towards the coast, where they break, scatter, and cause local mixing on the shelf. The forcing of regional models with remote internal tides enhances



the coastal internal tide energy (Siyanbola et al., 2023). Model simulations and satellite & in-situ observations have shown that internal tides feature temporal variability over various timescales (Rainville and Pinkel, 2006; Shriver et al., 2014; Zaron and Egbert, 2014; Ponte and Klein, 2015; Buijsman et al., 2017; Zaron, 2017; Nelson et al., 2019; Löb et al., 2020; Zhao and Qiu, 2023; Solano et al., 2023; Yadidya et al., 2024). Ultimately, this variability may contribute to the time variability in ocean mixing. In this study, we compare the seasonal variability in semidiurnal steric sea surface height (SSSH) with internal tide energetics in global ocean model simulations.

The seasonal variability in internal tides has been observed and simulated in numerous studies. Ray and Zaron (2011) noticed significant seasonal variations in internal tide sea surface height (SSH) in the northern South China Sea region using the in-phase component of the $M_2$ harmonic constant. The in-phase component of the internal tides has a constant phase and amplitude depending on the duration of the time series (Colosi and Munk, 2006; Ansong et al., 2015; Buijsman et al., 2020). Zaron (2019) showed global maps of annual modulates of $M_2$ baroclinic SSH using satellite altimeter data and observed that seasonal variations are observed in the Arabian Sea, the region between the Seychelles and Madagascar, the South China Sea, and the region offshore of the Amazon River plume. The annual modulations of the $M_2$ internal tides create a signal at $MA_2$ and $MB_2$ frequencies, where $MA_2$ is $M_2$ minus the annual frequency and $MB_2$ is $M_2$ plus the annual frequency (Huess and Andersen, 2001; Zaron, 2019). Zhao (2021) used 25 years of global satellite altimeter data and observed that seasonal phase variations are more dominant than seasonal amplitude variations for phase-locked internal tide SSH. Zhao (2021) also noticed strong seasonal variations in areas where a seasonal cycle in stratification is observed. Zhao and Qiu (2023) analyzed the seasonal variability of $M_2$ internal tides in the Luzon Strait. They suggested that ocean stratification and the Kuroshio current may be responsible for the seasonal variability.

Numerical model studies investigating the seasonal variability of the internal tide mainly focus on regional areas (Gerkema et al., 2004; Jan et al., 2008; Osborne et al., 2011; Zaron and Egbert, 2014; Yan et al., 2020). Gerkema et al. (2004) observed seasonal dependence in the generation and propagation of the simulated internal tides in the Bay of Biscay region. They found that the area-integrated barotropic to baroclinic conversion rate is 15% higher in summer compared to winter, which can be attributed to the seasonal thermocline. Zaron and Egbert (2014) showed around 10% of seasonality in the mode 1 phase speed of $M_2$ internal tides in a simulation centered on the Hawaiian Ridge.

Only a few studies have used global numerical models to identify seasonal variability of internal tides (Müller et al., 2012; Shriver et al., 2014). Müller et al. (2012) used the STORMTIDE Model to show differences in the phase-locked $M_2$ internal tide SSH amplitude for summer and winter months. They found a root mean square error of the amplitude differences between the summer and winter seasons exceeding five millimeters in the western Pacific, around Madagascar, and the Bay of Bengal. Shriver et al. (2014) utilized the global Hybrid Coordinate Ocean Model (HYCOM) to study the seasonal variability in phase-locked $M_2$ internal tide SSH by calculating the annual cycle in amplitude. They observed significant seasonal variability in the amplitude of $M_2$ internal tide in the Arabian Sea and the tropics.

The variability in the internal tide at the generation site can be due to seasonal fluctuations in barotropic tidal forcing (Liu et al., 2015) and stratification (Gerkema et al., 2004; Zhao, 2021; Schindelegger et al., 2022). Wind, stratification, or ice cover changes can impact the variability of barotropic tides over time (Kang et al., 2002; Müller et al., 2012; St. Laurent et al., 2008;





Bij de Vaate et al., 2021). Additionally, changes in stratification can affect the perturbation pressure, which, in turn, influences the rate of barotropic to baroclinic energy conversion. As the internal tides propagate, they can be influenced by the refraction of beams due to the temporal and spatial variability in eddies and stratification (Ponte and Klein, 2015; Duda et al., 2018), and dissipation (de Lavergne et al., 2019; Mukherjee et al., 2023).

    The understanding of the seasonal variability in internal tides in the global ocean has been limited by the short duration of
time series available from numerical experiments and the low spatial and temporal resolution of field measurements. While the sea surface height (SSH) of internal tides has been used in previous studies to explore seasonal changes, the seasonal variability of internal tide SSH and energetics has never been compared. This study aims to answer the following questions: (a) Which areas in the global ocean have high seasonal variability in semidiurnal internal tides? (b) How do the spatial and temporal variabilities in internal tide SSH and energetics from a global HYCOM simulation compare? (c) What explains
their differences? (d) What mechanisms cause the seasonal variability? To answer these questions, we analyze the seasonal variability in semidiurnal internal tides using two global HYCOM simulations with output durations of 5 years and 1 year. We will examine the seasonal trends in SSSH variance, barotropic to baroclinic conversion rate, kinetic energy (KE), available potential energy (APE), and pressure flux for semidiurnal internal tides.

    The rest of the paper is organized as follows: Section 2 explains the model simulation and the applied methodology. In
section 3, we compare the seasonal variability in the semidiurnal SSSH variance with that in the internal tide energetics. To confirm the accuracy of our findings, we also compare the seasonal variability in the HYCOM simulation with the satellite altimeter observation. Section 4 discusses the causes of the disparity in seasonal trends between SSSH variance and internal tide energetics. Finally, section 5 summarizes the study's key findings.

## 2   Model and Methodology

### 2.1   Model Simulations

This study uses two global non-data assimilative HYCOM simulations (expt 06.1 and expt 18.5) and an altimetry dataset. The list of datasets extracted from these simulations is given in Table 1. We use various data products of these simulations that already exist. For the validation of the HYCOM simulation (Buijsman et al., 2017; Buijsman et al., 2020), we use the altimetry dataset from Zhao (2021).

### 2.1.1   Expt 06.1

For expt 06.1, the data is available for one year, from October 2011 to September 2012. We use SSSH and internal tide energy terms from this simulation. This non-data assimilative simulation is forced with realistic atmospheric and tidal forcing ($M_2$, $S_2$, $N_2$, $K_1$, $O_1$). The horizontal resolution is 8 km with 41 vertical layers. In expt 06.1, an Augmented State Ensemble Kalman Filter (ASEnKF) technique (Ngodock et al., 2016) is applied to improve the accuracy of barotropic tides. A parameterized





**Table 1.** List of datasets used in this study.

| Model Name | Model/ Observation | Grid resolution [degrees] | Time series duration [years] | Products used in this study |
|---|---|---|---|---|
| Expt 06.1a | HYCOM | 1/12.5 | 1 (10/2011-09/2012) | Hourly modal pressure and baroclinic velocity amplitudes. Monthly-mean horizontal velocity eigenfunctions and buoyancy frequency values. Monthly-mean global barotropic to baroclinic conversion rate, KE, APE, and baroclinic flux. These data are extracted with methods discussed in Buijsman et al. (2020) and Raja et al. (2022). |
| Expt 06.1b | HYCOM | Subsampled at 6/12.5 | 1 (10/2011-09/2012) | Hourly time series of SSSH. |
| Expt 18.5 | HYCOM | Subsampled at 0.5 | 5 (01/2005-12/2009) | Hourly time series of SSSH |
| ZHAO21 | Altimetry | 0.2 | 25 (1992-2017) | Harmonic constants of $M_2$ for 4 seasons from Zhao (2021) |

topographic wave drag (Jayne and St. Laurent, 2001) and a scalar self-attraction and loading correction (SAL; Hendershott, 1972; Ray, 1998) are used. Buijsman et al. (2017) and Buijsman et al. (2020) have discussed this model simulation in detail.

We use hourly SSSH snapshots subsampled at $6/12.5°$ to analyze the seasonal variability in the semidiurnal internal tide SSSH. We refer to this time series as expt 06.1b (Table 1). The monthly-mean and depth-integrated semidiurnal barotropic to baroclinic conversion rate, KE, APE, and baroclinic flux fields are also computed by reconstructing harmonic timeseries for the sum of the $M_2$, $S_2$, and $N_2$ constituents. The SSSH amplitude ratios of $M_2$:$S_2$:$N_2$ are 1.0:0.44:0.30. Hence, our analysis for both SSSH and energetics focuses on the semidiurnal band, which is dominated by $M_2$ (Egbert and Ray, 2003). The modal pressure and baroclinic velocity amplitudes, and horizontal velocity eigenfunctions are used to calculate mode 1 baroclinic (BC) SSH and energy terms. We refer to these data as expt 06.1a (Table 1).

### 2.1.2 Expt 18.5

We also use 5-year-long datasets from a global HYCOM simulation with a horizontal resolution of 8 km and 32 vertical layers to analyze the seasonal variability. This simulation features realistic atmospheric and tidal forcing. It is forced with four semidiurnal constituents ($M_2$, $S_2$, $N_2$, and $K_2$) and four diurnal constituents ($K_1$, $O_1$, $P_1$, and $Q_1$). A parameterized topographic




wave drag (Arbic et al., 2010) and a scalar self-attraction and loading correction (SAL; Hendershott, 1972; Ray, 1998) are used. Shriver et al. (2012), Buijsman et al. (2016), and Nelson et al. (2019) have discussed this model simulation in detail.

To analyze the seasonal cycle in the semidiurnal internal tide SSSH, we use SSSH snapshots that are saved once per hour from 1 January 2005 to 31 December 2009. These snapshots are subsampled at $0.5°$ grid resolution.

## 2.2    Methodology

### 2.2.1    Seasonal variability in steric sea surface height

We analyze the seasonal variability in the semidiurnal SSSH. Steric SSH is computed inline during the HYCOM simulation
(Savage et al., 2017). Although it has been shown in Kaur (2024) and Zaron and Ray (2023) that the semidiurnal internal tide amplitude of SSSH is larger than the true semidiurnal internal tide SSH amplitude by about 20%, Kaur (2024) shows that the spatio-temporal variability is the same for both SSH metrics. We use a harmonic analysis to extract the $M_2$, $S_2$, and $N_2$ constituents, from which we calculate the semidiurnal signal. To extract the semidiurnal signal, we prefer the harmonic analysis over the bandpass analysis because the latter method also captures some mesoscale variance, particularly at higher latitudes
(results not shown), and numerical noise (thermobaric instability; Buijsman et al., 2020).

We extract monthly $M_2$, $S_2$, and $N_2$ amplitudes using the five-year hourly time series of SSSH from expt 18.5. Using a least-squares fit analysis, we compute the harmonic constants of the $M_2$, $S_2$, $N_2$, $K_1$, and $O_1$ constituents for each month (730 hours). The duration of 730 hours is long enough to resolve these five constituents. However, there is a possibility of tidal aliasing due to $K_2$ and $P_1$. This is because $K_2$ and $S_2$, and $K_1$ and $P_1$ both are separated by 2 cycles per year. It is important
to note that this tidal aliasing only affects the semiannual seasonal signal for $S_2$, and not the annual signal (results not shown). The SSSH time series of the $M_2$, $S_2$, and $N_2$ internal tide can be written as

$$\eta_{i,m} = \sum_j a_j(m)\cos(\omega_j t_i) + b_j(m)\sin(\omega_j t_i), \tag{1}$$

where $i = 1, ..., I$, $I$ is the number of hours in the month $m$, $a$ and $b$ are harmonic constants for the month and $j$ refers to the $M_2$, $S_2$, $N_2$ constituents, $t$ represents time, and $\omega$ is the frequency. The semidiurnal SSSH variance for each month is calculated as

$$\sigma_{D2}^2(m) = \frac{1}{I}\sum_{i=1}^I (\eta_{i,m})^2. \tag{2}$$

To calculate the seasonal variability in the semidiurnal internal tides SSSH variance, the annual cycle is fitted to the 5-year time series of the monthly semidiurnal SSSH variance ($\sigma_{D2}^2$) with a least-squares method after removing the linear trend

$$\hat{\sigma}_{D2}^2(m) = A_a\cos(\omega_a t_m - \phi_a), \tag{3}$$

where $\omega_a$ is the annual frequency, $A_a$ and $\phi_a$ are the amplitude and phase of the annual signal, respectively, $\hat{\sigma}_{D2}^2$ is the fitted time series, $t_m = 30.42m$, $m$ is the index for each month, which has 30.42 days.





We calculate the coefficient of determination ($R^2$) for the annual fit of the time series of the monthly variance. It identifies
how much of the variability in the semidiurnal SSSH variance is due to the seasonality. The coefficient of determination for the
fit is given by

$$R^2 = 1 - \frac{\sum_{m=1}^{M}[\sigma_{D2}^{2\prime}(m) - \hat{\sigma}_{D2}^{2}(m)]^2}{\sum_{m=1}^{M}[\sigma_{D2}^{2\prime}(m)]^2},$$ (4)

where $\sigma_{D2}^{2\prime}(m)$ is the detrended time series of the monthly semidiurnal variance, and $M$ is the total number of months.

### 2.2.2 Internal tide energetics

We analyze the seasonal variability in the semidiurnal internal tide energetics and compare it with the SSSH variance. Follow-
ing Buijsman et al. (2020), we compute the monthly-mean and depth-integrated semidiurnal barotropic to baroclinic energy
conversion rate, KE, APE, and baroclinic energy flux from hourly 3D data of expt 06.1a. The semidiurnal band includes the
$M_2$, $S_2$, and $N_2$ constituents.

The depth-integrated and time-averaged internal tide energy balance equation is written as (Buijsman et al., 2016; Kang and
Fringer, 2012)

$$\langle C \rangle = \langle \frac{\partial E}{\partial t} \rangle + \langle \nabla_h \cdot \mathbf{F} \rangle + \mathcal{R},$$ (5)

where $\langle \rangle$ indicates the time-average over a month, $C$ is the depth-integrated barotropic to baroclinic conversion, $E$ is the
depth-integrated total baroclinic wave energy, $\nabla_h \cdot \mathbf{F}$ is the horizontal divergence of the depth-integrated baroclinic energy flux
$\mathbf{F} = (F_x, F_y)$, $\mathcal{R}$ is the residual, which is mostly due to baroclinic energy dissipation, and $\frac{\partial E}{\partial t}$ is the tendency term, which is
$\approx 0$ after time averaging (Buijsman et al., 2016; Buijsman et al., 2020).

The depth-integrated and time-averaged conversion of baroclinic tides from barotropic tides for each $x, y$ coordinate is

$$\langle C \rangle = \frac{1}{T} \int_0^T W(z=-H,t) p'(z=-H,t) dt,$$ (6)

where $W(z=-H,t)$ is the vertical barotropic velocity at the sea floor, $p'(z=-H,t)$ is the perturbation pressure at the sea
floor, $z$ is the vertical coordinate, $T$ is the number of hours in the month, and $H$ is the water depth. The perturbation pressure
is calculated as in Nash et al. (2005) and by removing the depth-mean pressure. The vertical barotropic velocity at the sea floor
for each $x, y$ coordinate is given by

$$W(z=-H,t) = -\mathbf{U}(t) \cdot \nabla_h H,$$ (7)

where $\mathbf{U} = (U,V)$ are the barotropic velocities in the $x$ and $y$ directions. The depth-integrated and time-averaged baroclinic
flux for each $x, y$ coordinate is calculated as

$$\langle \mathbf{F} \rangle = \frac{1}{T} \int_0^T \int_{-H}^0 \mathbf{u}'(z,t) p'(z,t) dz dt,$$ (8)




where $\mathbf{u}' = (u', v')$ are the horizontal baroclinic velocities in the $x$ and $y$ directions. The depth-integrated and time-averaged KE for each $x$, $y$ coordinate is calculated as

$$\langle KE \rangle = \frac{1}{T} \int\limits_{0}^{T} \int\limits_{-H}^{0} \frac{1}{2}\rho_0 (u'(z,t)^2 + v'(z,t)^2)dzdt, \tag{9}$$

where $\rho_0$ is the reference density. The depth-integrated and time-averaged APE for each $x$, $y$ coordinate is calculated as

$$\langle APE \rangle = \frac{1}{T} \int\limits_{0}^{T} \int\limits_{-H}^{0} \frac{g^2 \rho'(z,t)^2}{2\rho_0 N^2(z,t)} dzdt, \tag{10}$$

where $g$ is the gravitational acceleration, $N$ is the buoyancy frequency and $\rho'$ is the perturbation density. The depth-integrated total baroclinic energy is the sum of depth-integrated KE and APE. The buoyancy frequency is calculated as

$$N^2 = -\frac{g}{\rho_0} \frac{\partial \langle \rho \rangle}{\partial z}, \tag{11}$$

where $\langle \rho \rangle$ is the mean potential density and $(\frac{\partial \langle \rho \rangle}{\partial z})$ is the vertical density gradient. In the remainder of the paper, we will drop the $\langle \rangle$ when discussing the time-averaged energy terms.

For expt 06.1, all variables (SSSH variance, barotropic to baroclinic conversion, KE, APE, total energy, and flux) are calculated for each month from October 2011 to September 2012. The exact number of hours per month is used. The hours for each month from October 2011 to September 2012 are as follows: 744, 720, 744, 744, 696, 744, 720, 744, 720, 744, 383, and 889. For August, we use the first two weeks of data because the model data for the third week is missing. We add the fourth week of data from August to September, resulting in 5 weeks of data for September. However, the short months, February and August, show outlier values. Therefore, we remove the monthly-mean values for February and August for each grid point and then linearly interpolate the values for these two months using data from adjacent months. The SSSH variance values are subsampled at $6/12.5°$ resolution. Hence, we also subsample KE, APE, total energy, and flux to $6/12.5°$ resolution.

### 2.2.3 Modal energetics

We decompose internal tide SSH and energetics into vertical modes to better understand the discrepancies in seasonal trends between SSSH variance and energy terms. For the calculation of mode 1 baroclinic SSH, KE, and APE, 3D HYCOM fields from expt 06.1a are decomposed into vertical modes following Buijsman et al. (2020) and Raja et al. (2022). The eigenfunctions of the first 5 modes are computed for each month by solving the Stürm Liouville equation using the monthly-mean and spatially varying buoyancy frequency (Gerkema and Zimmerman, 2008). The horizontal velocity eigenfunctions are projected onto the perturbation pressure and baroclinic velocity time series to obtain a modal amplitude time series for pressure and velocities.

To compare with SSSH, we compute the mode 1 SSH. For each $x$, $y$ coordinate applies

$$p_1(z,t) = \tilde{p}_1(t)\mathcal{U}_1(z), \tag{12}$$





where $p_1$ is the mode 1 pressure, $\tilde{p}_1(t)$ is the mode 1 amplitude time series of pressure, and $\mathcal{U}_1(z)$ is the horizontal velocity

eigenfunction of mode 1. The mode 1 SSH is calculated as

$$\eta_1(t) = \frac{p_1(z=0,t)}{g\rho_0}. \tag{13}$$

We extract the harmonic constants for the $M_2$, $S_2$, and $N_2$ internal tide from the hourly time series of mode 1 SSH and bottom

perturbation pressure ($p_1(z=-H,t)$) for each month. Then, the variance for semidiurnal mode 1 SSH and bottom perturbation

pressure is calculated using Eqs. (1) and (2).

For calculation of the mode 1 semidiurnal KE and APE, we extract the harmonic constants for $M_2$, $S_2$, and $N_2$ constituents

for each month from the time series of the mode 1 amplitudes of baroclinic velocities and pressure. The mode 1 semidiurnal

KE and APE are calculated as (Kelly et al., 2012)

$$KE_1 = \sum_j (|\hat{u}_{1j}|^2 + |\hat{v}_{1j}|^2)\frac{H\rho_0}{4}, \tag{14}$$

$$APE_1 = \sum_j (1 - \frac{f^2}{\omega_j^2})\frac{|\hat{p}_{1j}|^2}{c_1^2\rho_0}\frac{H}{4}, \tag{15}$$

where $\hat{u}_{1j}$, $\hat{v}_{1j}$, and $\hat{p}_{1j}$ are the mode 1 complex harmonic constants of the baroclinic velocities and perturbation pressure for

constituent $j$, $c_1$ is the mode 1 eigenspeed, and $f$ is the Coriolis frequency.

The mode 1 variables are also linearly interpolated for February and August for each grid point using the same methodology

as we employed for the undecomposed fields. Additionally, we subsample these variables to $6/12.5°$ resolution to compare

with SSSH variance.

### 2.2.4   Satellite altimeter data

We validate the seasonal variability in the HYCOM mode 1 $M_2$ baroclinic SSH variance and KE with the satellite altimeter

data of Zhao (2021). Zhao (2021) could only extract mode 1 $M_2$ amplitudes with reasonable accuracy for 4 seasons: winter

(January, February, and March), spring (April, May, and June), summer (July, August, and September), and fall (October,

November, and December). The variance for each season is calculated as

$$\sigma_s^2 = \frac{A_s{}^2}{2}, \tag{16}$$

where $A_s$ is the mode 1 $M_2$ internal tide SSH amplitude for season $s$.

## 3   Results

### 3.1   Spatial variability in SSSH and energy

The mean over 12 months for semidiurnal SSSH variance, depth-integrated semidiurnal barotropic to baroclinic conversion

rate, depth-integrated semidiurnal baroclinic energy, and depth-integrated semidiurnal baroclinic flux for expt 06.1 are shown

in Figure 1. The baroclinic energy and flux values are subsampled at $6/12.5°$ to compare with the SSSH variance.







**Figure 1.** (a) Monthly semidiurnal SSSH variance, (b) depth-integrated semidiurnal barotropic to baroclinic conversion rate, (c) baroclinic semidiurnal energy, and (d) baroclinic semidiurnal flux averaged over 12 months. In (a), (c), and (d), black bathymetry contours are plotted at 0 m and 2000 m. In (a), the following regions are marked: (1) East of Philippines, (2) Hawaii, (3) Tropical SW Pacific, (4) Tropical South Pacific, (5) Georges Bank, (6) Amazon Plume, (7) Madagascar, and (8) Arabian Sea. In (b), conversion rates are area-averaged to 1° grid resolution, and black bathymetry contours are plotted at 0 m. In (c), the gray polygon marks the area affected by thermobaric instabilities.





The barotropic to baroclinic conversion rate in Figure 1b is higher at ridges and rough topography, highlighting the internal tide generation sites. SSSH variance, energy, and flux values are also higher near the generation sites (Figure 1). The patterns for APE and KE (not shown) are the same as total energy (Figure 1c). The polygon in Figure 1c marks an area in the North Pacific with elevated KE, which may be attributed to thermobaric instabilities (TBI). TBI is a numerical instability present in Lagrangian/isopycnic vertical coordinate ocean models due to inaccurate compensation for compressibility in calculating

pressure gradient accelerations (Hallberg, 2005). TBI is known to occur in the North Pacific Ocean (Buijsman et al., 2016; Buijsman et al., 2020). Although TBI is a broadband signal with a variable phase, it is possible that some of this noise projects on the harmonic constants. We omit the area with TBI because it adversely impacts the seasonal variability analysis.

### 3.2    Seasonal variability in steric sea surface height

The first objective is to analyze the seasonal variability in semidiurnal internal tide SSSH variance. We use the detrended

monthly variance ($\sigma^{2\prime}_{D2}$) of the combined $M_2$, $S_2$, and $N_2$ internal tides, which is computed with hourly time series of SSSH from expt 18.5. We use expt 18.5 because this simulation has a 5-year duration, which benefits the calculation of the annual seasonal cycle. Using a least-squares fit analysis, we calculate the annual cycle in the monthly variance of semidiurnal internal tide SSSH. The amplitude of the annual cycle is normalized by the mean semidiurnal variance over 60 months (Figure 2a). We are using the normalized amplitude (Figure 2a) and $R^2$ (Figure 2c) as indicators for the seasonal variability. If the normalized

amplitude is small, the variance does not vary much seasonally. Hence, we focus on areas where both the normalized amplitude and $R^2$ are significant.

The internal tides generated in the coastal areas of Georges Bank and the Arabian Sea display the largest seasonal variability (Figure 2a and c). Figure 2b shows that the seasonal cycle of the semidiurnal internal tide SSSH variance in the Northern Hemisphere is 180° out of phase with the variance in the Southern Hemisphere. This indicates that seasonal changes in the

semidiurnal SSSH variance may be due to stratification, which can affect the propagation and/or generation of the internal tides. In the next section, we investigate whether the same seasonal variability is present in the internal tide energetics.

### 3.3    Seasonal variability in internal tide energetics

In this section, we analyze the seasonal trend in semidiurnal internal tide energetics and compare it with the seasonal trend in semidiurnal SSSH variance. To do so, we use one-year data from expt 06.1 to calculate the monthly semidiurnal SSSH

variance, depth-integrated conversion rate, KE, APE, energy, and flux for $M_2$, $S_2$, and $N_2$ constituents. Unfortunately, the annual cycle fit similar to Figure 2 for the internal tide energetics is very noisy because one year of data is insufficient to fit the annual signal accurately. Hence, we do not show these results.

To better visualize the seasonal trends, we zonally average the conversion rate, flux, SSSH variance, KE, APE, and total energy over 10-degree latitude bins for the Atlantic and the Pacific Oceans for each one-month period. The values in areas

shallower than 100 m are excluded from the analysis. This is because the model does not resolve internal tides satisfactorily in these areas. To derive the anomaly time series for these variables, we remove and normalize by their annual-mean values. The anomaly plots are presented in Figures 3 and 4 for the Pacific and Atlantic Oceans, respectively.





**Figure 2.** (a) The normalized amplitude and (b) phase of the annual cycle of the monthly semidiurnal SSSH variance ($\sigma_{D2}^{2\prime}$). (c) The coefficient of determination ($R^2$) for the fit. Black bathymetry contours are plotted at 0 and 2000 m. The normalized amplitude, phase, and $R^2$ of the fit are computed for each point and smoothed by taking a 9-point (3x3 square box) running mean. The areas with small internal tides (mean monthly variance < 0.01 cm$^2$) are removed.



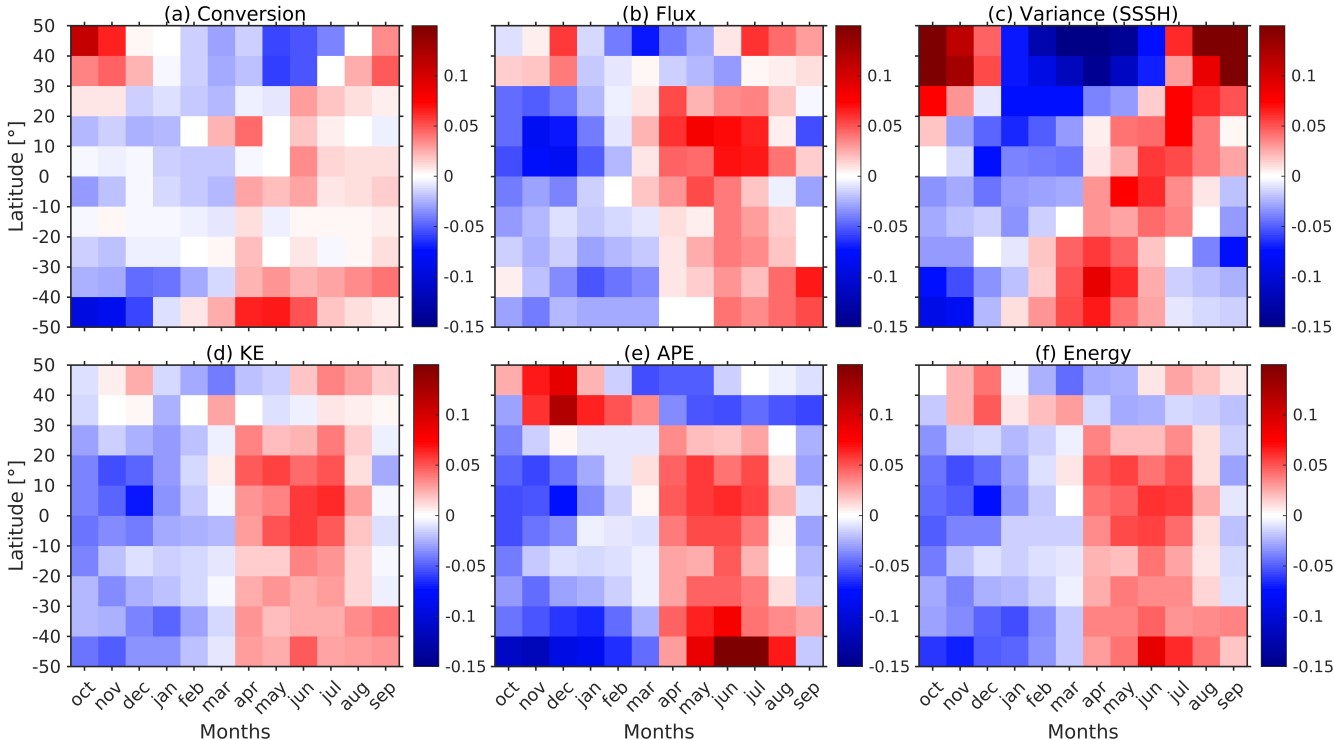

**Figure 3.** Zonally averaged anomaly time series of monthly semidiurnal (a) barotropic to baroclinic conversion rate, (b) baroclinic energy flux, (c) SSSH variance, (d) KE, (e) APE, and (f) total energy for the Pacific Ocean. The anomalies are computed by removing and normalizing by the annual-mean values.

A seasonal cycle is observed in all variables (conversion rate, flux, SSSH variance, KE, APE, and total energy) in both the Pacific and Atlantic oceans (Figures 3 and 4). While the seasonal trends are broadly similar for the energy terms in both oceans, they differ from the trends in the SSSH variance. Similar to Figure 2b, the seasonal cycle of the SSSH variance is out of phase in the Northern and Southern Hemispheres in Figures 3 and 4. In contrast, the energy terms do not follow the same trends for the two hemispheres. The seasonal cycles of the energy terms are out of phase with SSSH variance, and no apparent differences are present between the Northern and Southern Hemispheres. However, both SSSH and energetics tend to show the largest amplitudes at higher latitudes.

The amplitude of the seasonal cycles in Figures 3 and 4 is maximally about 15% of the annual-mean value. The percentage change is higher for the SSSH variance as compared to the conversion rate and KE. The percentage change for APE is also larger than the conversion rate and KE at higher latitudes. For a free-propagating internal tide, $\frac{KE}{APE} = \frac{\omega^2 + f^2}{\omega^2 - f^2}$ (Alford and Zhao, 2007). As $\omega \approx f$ at higher latitudes, APE tends to 0. Therefore, APE is small at higher latitudes, and the percentage change for APE is large. To determine that the normalization is not misrepresenting the seasonal signal, we show the non-normalized anomalies in Figures A1 and A2 in Appendix A. These figures show that the seasonal trends remain consistent. However, there





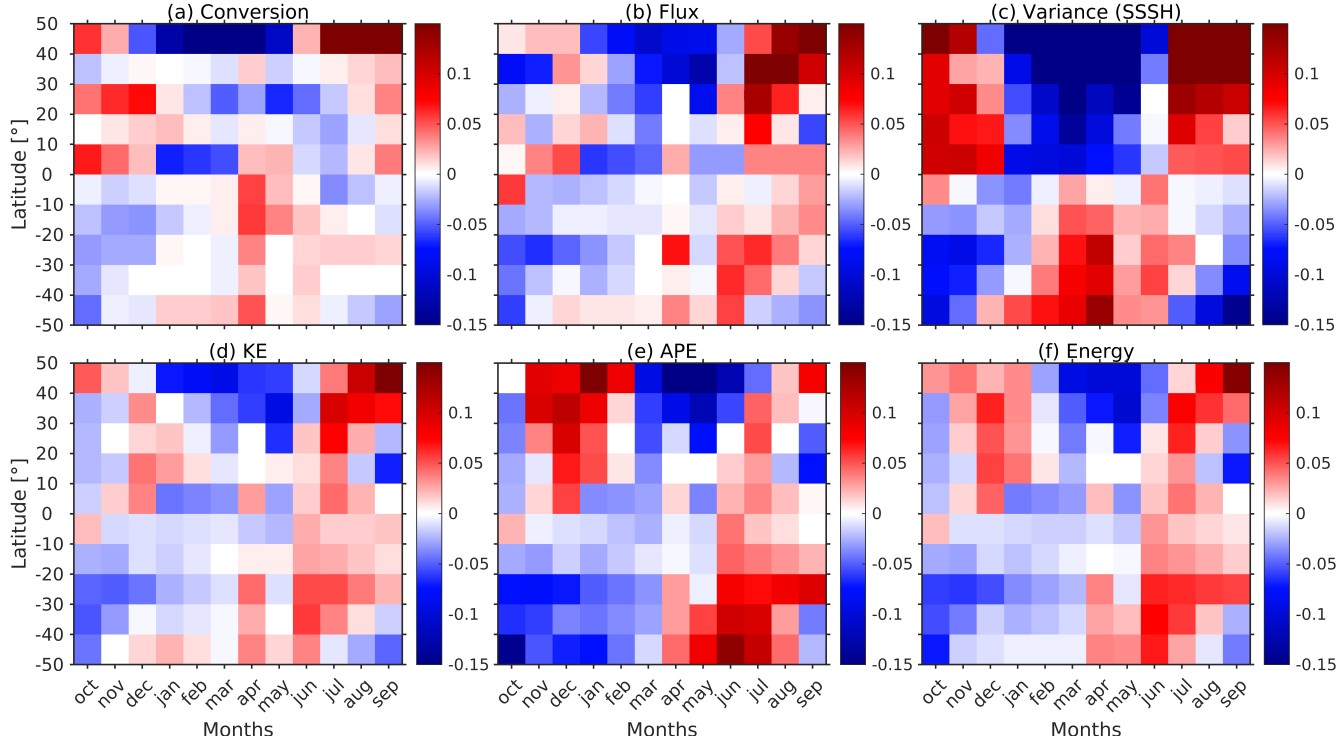

**Figure 4.** The same as Figure 3, but for the Atlantic Ocean.

is a difference in the amplitude because if the internal tide signal is small (large), it can increase (decrease) the percentage change. Additionally, it is important to emphasize that this study focuses primarily on trends and percentage change; therefore, utilizing the normalized data is preferable.

In the Pacific Ocean, the seasonal signal for the SSSH variance in the tropical region, spanning from -20° to 20°, is marginally
out of phase when compared to the polar regions (Figure 3c). In this region, the seasonal signal in the SSSH variance is in sync with the internal tide energetics. In the Atlantic Ocean (Figure 4), the seasonal cycle for energy terms is noisier than in the Pacific Ocean. This may be because the internal tide generation is weaker in the Atlantic Ocean, with most of the generation happening at deep ridges. The strong seasonal signal in the conversion between 40°N to 50°N in Figure 4a (and A2a) is attributed to Georges Bank.

To further compare the seasonal signals, we plot in Figure 5 the area-averaged semidiurnal SSSH variance, KE, and conversion anomalies for each month for eight regions (red boxes in Figure 1a). The seasonal signal of KE and conversion is similar to that of the SSSH variance for Georges Bank and the Arabian Sea, where the strongest seasonal variability in internal tides is observed (Figure 5i, j, o, and p). The strong seasonal variability in conversion rate in the Arabian Sea and Georges Bank and their relation to the stratification are further explained in Appendix B. The seasonal amplitudes of SSSH variance, KE,
and conversion at Georges Bank and the Arabian Sea are ∼50% and ∼20%, respectively. The correlation coefficient between





**Figure 5.** Area-averaged (left column) monthly semidiurnal SSSH variance (blue line) and KE (orange line) and (right column) normalized anomaly time series of monthly semidiurnal SSSH variance (blue line), KE (orange line), and conversion (black line) for regions marked by the red square boxes in Figure 1a. C1 and C2 are the correlation coefficients between SSSH variance and conversion, and between KE and conversion, respectively.





SSSH variance and conversion at Georges Bank and the Arabian Sea is 0.99 and 0.90, respectively. However, for the Tropical SW Pacific and the Tropical South Pacific (Figure 5e, f, g, and h), the seasonal signal of KE is similar to the SSSH variance but with a phase lag. For other regions, the discrepancies are more complex. Overall, the conversion shows a better correlation with KE than SSSH variance, except east of the Philippines, where the correlation coefficient between KE and SSSH variance (0.78) is higher than between KE and conversion (0.23).

Our analysis indicates that the conversion rate is the primary factor responsible for the seasonal variability in internal tide energetics because the seasonal trends of conversion and KE are similar. However, the seasonal trends of SSSH variance are different from those of conversion and energy terms, except for Georges Bank and the Arabian Sea, where the seasonal variability is strong. In the next section, we will validate the seasonal variability in HYCOM with altimetry to ensure that the seasonal variability in HYCOM sea surface height is realistic. In the discussion section, we will explain the observed difference in the seasonal variability between SSSH variance and internal tide energetics in other regions.

### 3.4 Comparison with the satellite altimeter data

To validate the seasonal variability in our model, we compare the HYCOM output with the satellite altimeter data from Zhao (2021). Zhao (2021) analyzed the seasonal variability of the mode 1 $M_2$ internal tide using satellite altimeter data. We compare the mode 1 $M_2$ internal tide variance from the satellite altimeter data for each season with the mode 1 $M_2$ SSH variance and mode 1 $M_2$ KE from the HYCOM simulation expt 06.1a in Figure 6. It is discussed and shown later that semidiurnal SSSH and mode 1 SSH variance are in good agreement. To ensure accuracy, we omit areas with strong mesoscale activity from the satellite altimeter and HYCOM data. Additionally, we interpolate the mode 1 $M_2$ SSH variance and KE from the HYCOM simulation to the same locations as the altimetry data for each season. We then zonally average the $M_2$ variance from the satellite altimeter data, the $M_2$ baroclinic SSH variance, and the KE over $10°$ latitude bins for the Atlantic and the Pacific Oceans for each season, while also removing regions of weak internal tides (As in Zhao (2021), areas with $M_2$ internal tide SSH amplitude $< 0.2$ cm are removed) for all variables. Finally, we remove and normalize by the annual-mean. The results are presented in Figure 6.

The seasonal variability of both HYCOM and satellite altimeter $M_2$ SSH variance for the Pacific and Atlantic Oceans are in reasonable agreement. However, the level of agreement between the two is stronger in the Atlantic Ocean than in the Pacific Ocean. Moreover, the seasonal SSH variance of the satellite altimeter data is noisier than the HYCOM SSH variance. The reasons for this are not clear to us. It could be attributed to the sparseness of the satellite altimeter data in time and space. Despite these discrepancies, the trends observed in KE are different from both the HYCOM and satellite altimeter SSH variance, indicating that the seasonal variability in KE is different from the internal tide SSH variability. This comparison suggests that the trends in baroclinic SSH variance in HYCOM are realistic but that they do not reflect the seasonal variability of the energy in the internal tide except in Georges Bank and the Arabian Sea (Figure 5).





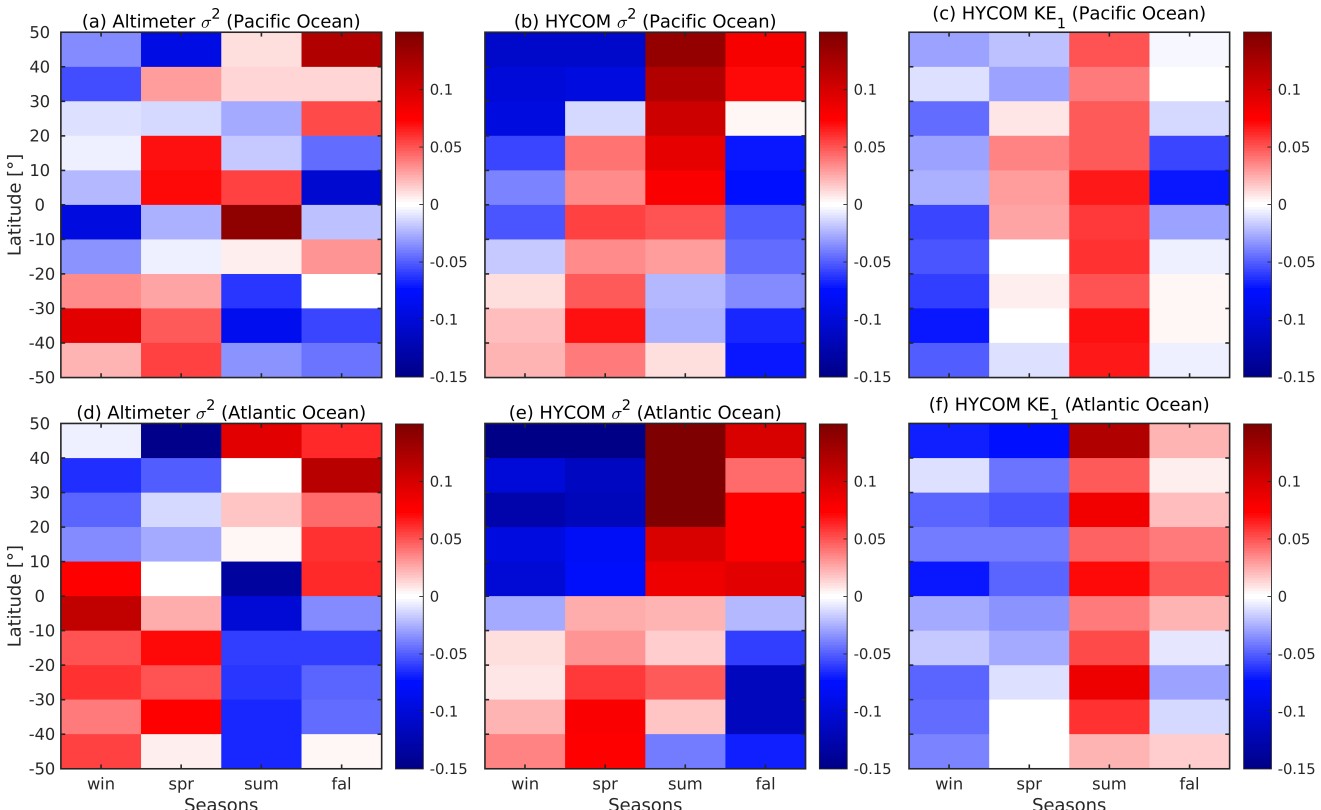

**Figure 6.** Zonally averaged normalized anomaly time series of seasonal-mean mode 1 $M_2$ internal tide SSH variance for (a and d) the satellite altimeter data and (b and e) the HYCOM data, and (c and f) mode 1 $M_2$ KE from HYCOM. The anomaly time series for the Pacific Ocean are in (a-c) and for the Atlantic Ocean are in (d-f).

## 4 Discussion

In this section, we explore the causes of the differences in the seasonal variability between SSSH variance and the energy terms. SSSH is strongly affected by the density of the surface layers, which varies significantly due to seasonal temperature changes

(Qu and Melnichenko, 2023). Based on our analysis, the seasonal changes in semidiurnal SSSH variance may not accurately represent the actual seasonal changes in internal tide energy because the internal tide SSH may be modulated by changes in surface temperature. To understand this better, we compare the seasonal variability in SSSH variance to mode 1 SSH variance, KE, APE, bottom perturbation pressure variance, and depth-mean buoyancy frequency.

We compute the mode 1 semidiurnal baroclinic SSH variance, bottom perturbation pressure variance, KE, and APE for

each month using 3D fields from expt 06.1a. These variables are based on reconstructed time series for the $M_2$, $S_2$, and $N_2$ constituents. We consider mode 1 because the internal tide SSSH is dominated by mode 1 (Zhao et al., 2019; Buijsman et al.,



2020). The spatial patterns of the time-mean bottom perturbation pressure variance are similar to the time mean semidiurnal SSSH variance in Figure 1a and is not shown.

We compare the seasonal trends in semidiurnal SSSH variance, mode 1 semidiurnal baroclinic SSH variance, KE, APE,
bottom perturbation pressure variance, and $N^2$ for the Pacific and Atlantic Oceans in Figures 7 and 8. We zonally average these variables over 10-degree latitude bins for each basin and one-month segment. For all variables, shallow areas are removed (depth < 100 m). To derive the anomaly time series, we remove and normalize by the annual-mean values. We consider the seasonal variability in the bottom perturbation pressure variance to better understand the effect of surface stratification on the SSSH variance. As expected, the undecomposed SSSH and mode 1 SSH variance have identical spatial and seasonal trends in
Figures 7b, c and 8b, c. In contrast, for both the Atlantic and Pacific Oceans, the seasonal variability in bottom perturbation pressure variance resembles that of the energy terms and not that of the mode 1 SSH variance, which is based on the surface perturbation pressure. Interestingly, the seasonal trend in the depth-mean buoyancy frequency is similar to the SSSH variance and mode 1 SSH variance, but it is 1-2 months ahead of the SSSH variance. The seasonal signal in the buoyancy frequency is out of phase in the Northern and Southern Hemispheres, which is similar to what we observe for the SSSH variance.

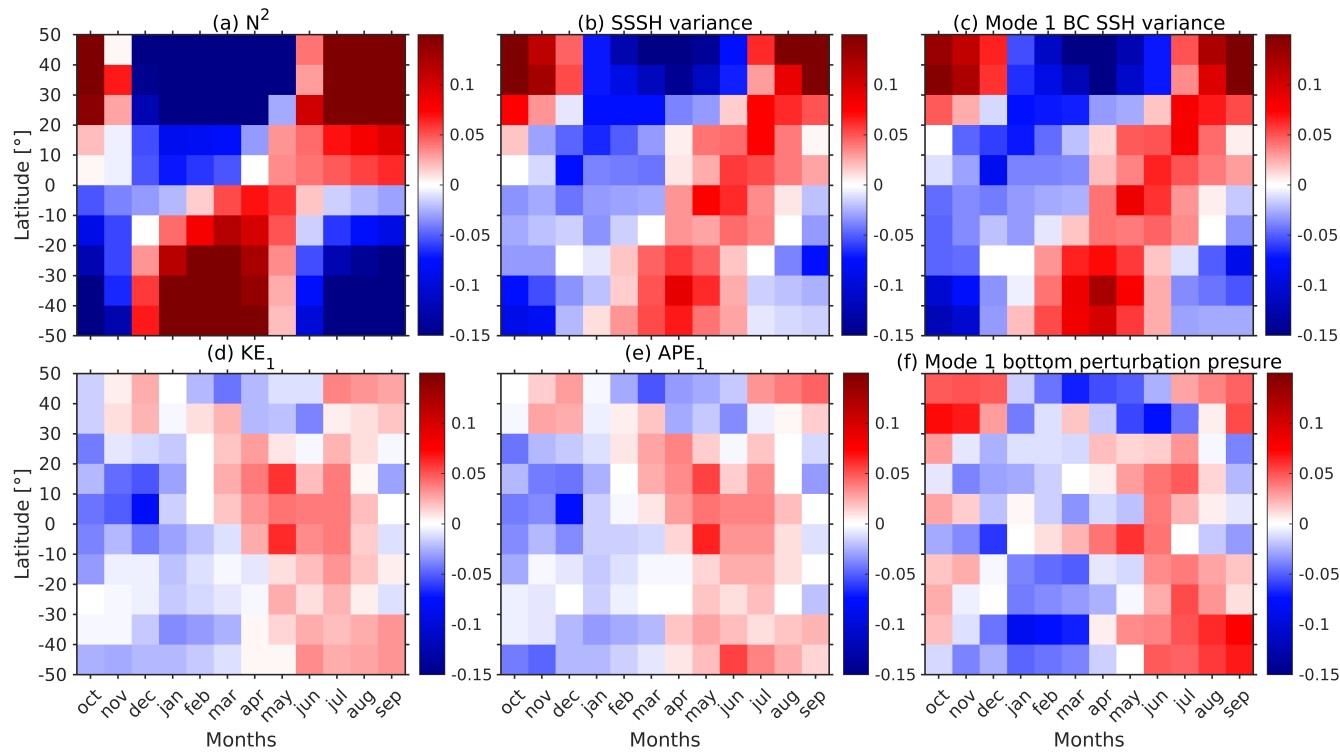

**Figure 7.** Zonally averaged normalized anomaly time series of monthly-mean (a) depth-mean $N^2$ and semidiurnal (b) undecomposed SSSH variance, (c) mode 1 SSH variance, (d) mode 1 KE, (e) mode 1 APE, and (f) mode 1 bottom perturbation pressure variance for the Pacific Ocean. The anomalies are computed by removing and normalizing by the annual-mean values.



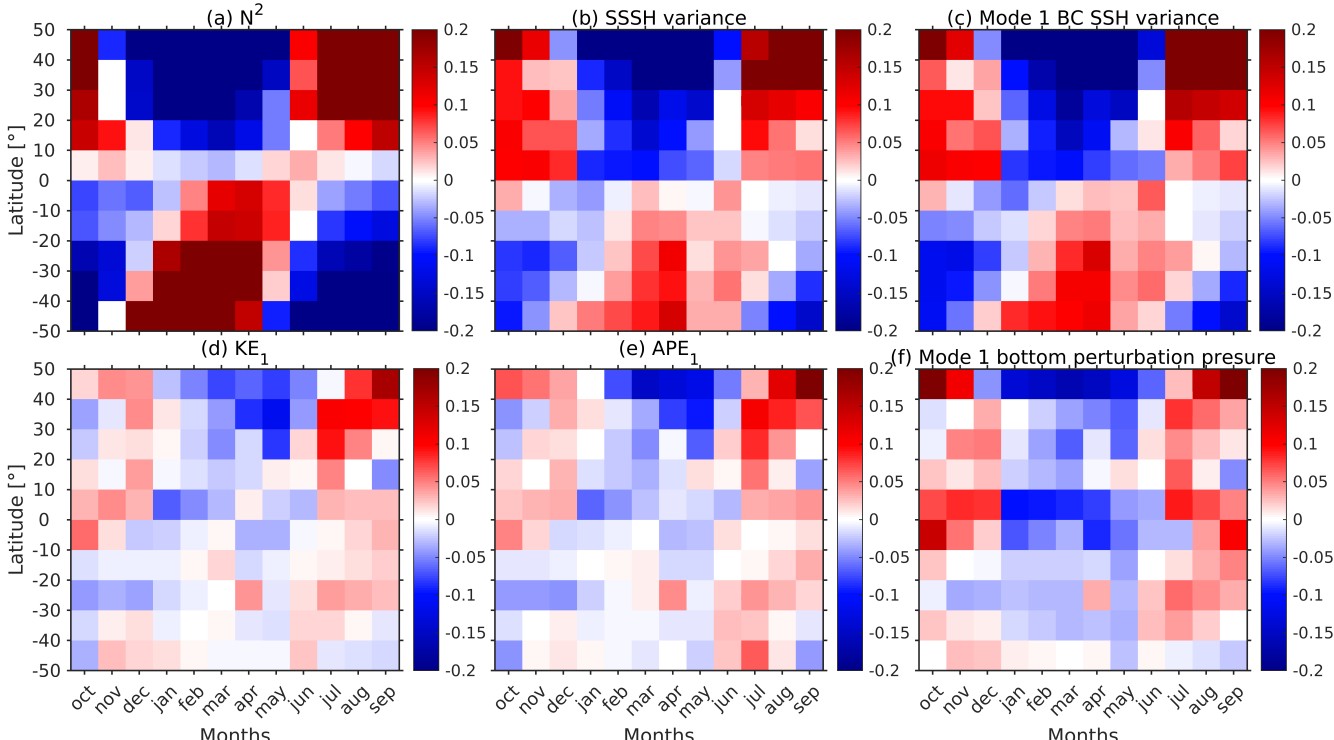

**Figure 8.** The same as Figure 7, but for the Atlantic Ocean.

The mode 1 SSH is computed as $\frac{\tilde{p_1}(t)\mathcal{U}_1(z=0)}{g\rho_0}$. To understand what is modulating the semidiurnal mode 1 SSH variance, we study the seasonal trend in mode 1 horizontal velocity eigenfunction at $z=0$ and the variance of the semidiurnal mode 1 perturbation pressure amplitude for the Pacific and Atlantic Oceans (Figure 9). The seasonal trends in $\mathcal{U}_1^2(z=0)$ are similar to the mode 1 SSH variance. The area-averaged correlation coefficient between the mode 1 SSH variance and $\mathcal{U}_1^2(z=0)$ is 0.84 and 0.89 for the Pacific and Atlantic Oceans, respectively. Moreover, the seasonal variability of $\mathcal{U}_1^2(z=0)$ is similar to

the depth-averaged buoyancy frequency anomaly in Figures 7a and 8a. Specifically, when the buoyancy frequency is surface intensified at the end of summer, $\mathcal{U}_1^2$ is also surface intensified. Therefore, we conclude that the surface density stratification is the main factor that modulates the seasonal variability in semidiurnal SSSH variance. In contrast, the variance of the semidiurnal mode 1 perturbation pressure amplitude ($\tilde{p}_1$) in Figure 9c and f is more in agreement with the mode 1 KE and APE variability in Figures 7 and 8. We note that $\mathcal{U}_1(z)$ does not contribute to the depth-integrated monthly values of mode 1 KE

and APE because of the normalization condition ($\frac{1}{H}\int_{-H}^0 \mathcal{U}_1^2(z)dz = 1$; Buijsman et al., 2020). Hence, the seasonal effect due to stratification observed for surface values of $\mathcal{U}_1$ disappears when depth-integrating.

Alternatively, we can also explain the modulation by considering APE. If we assume that the barotropic-to-baroclinic conversion rate remains constant throughout the year, we can assume that APE is constant. APE is proportional to $\frac{\rho'(z,t)^2}{N^2(z,t)}$ (Eq.





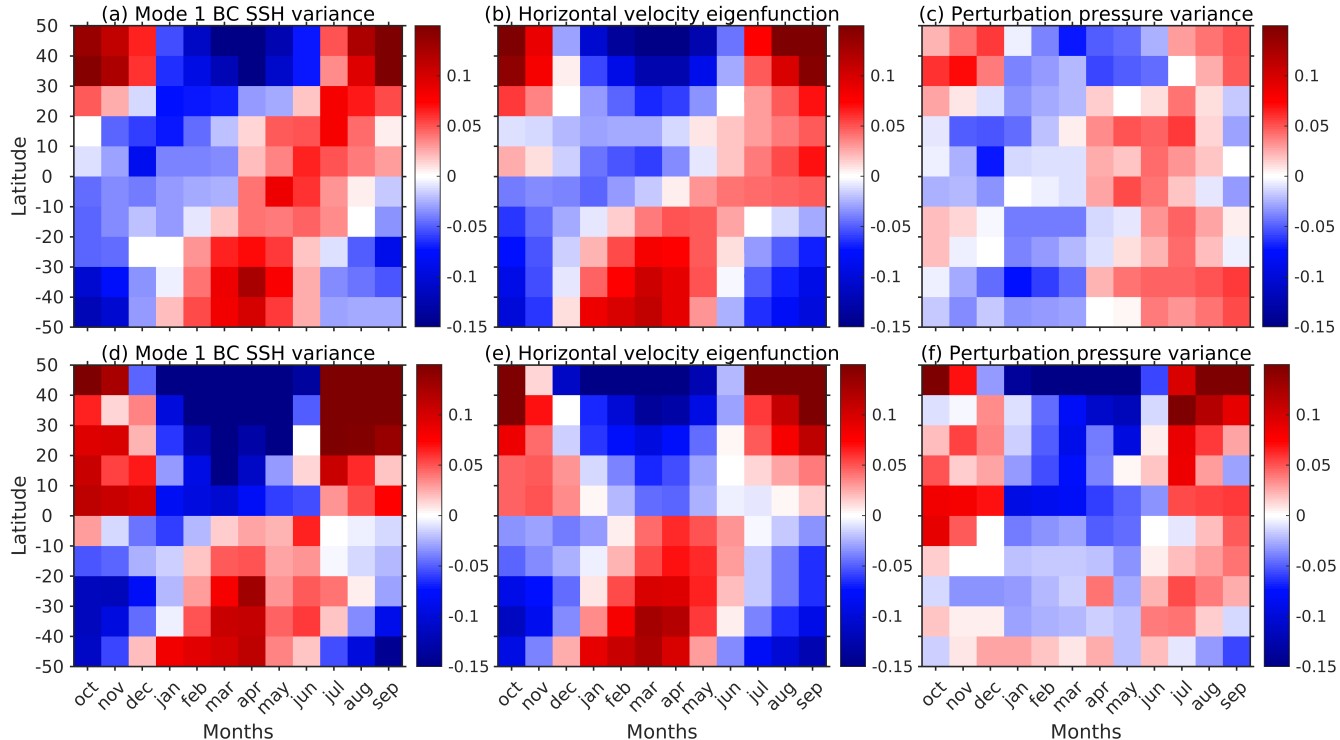

**Figure 9.** Zonally averaged normalized anomaly time series of (a and d) semidiurnal mode 1 baroclinic SSH variance, (b and e) mode 1 horizontal velocity eigenfunction ($\mathcal{U}_1^2$) at $z = 0$, and (c and f) semidiurnal mode 1 perturbation pressure amplitude ($\tilde{p_1}$) variance for the Pacific Ocean (top row) and Atlantic Ocean (bottom row).

10). If there is an increase (decrease) in surface temperature, $N$ also increases (decreases), which means that $\rho'(z,t)$ will also

increase (decrease) for APE to remain constant. This increase (decrease) in $\rho'(z,t)$ results in an increase (decrease) in SSSH.

## 5   Conclusions

In this study, we compare the seasonal variability of semidiurnal steric sea surface height (SSSH) with internal tide energetics, which are extracted from two non-data assimilative global Hybrid Coordinate Ocean Model (HYCOM) simulations. We analyze the seasonal trends in SSSH variance, barotropic to baroclinic conversion rate, kinetic energy (KE), available poten-

tial energy (APE), and pressure flux for semidiurnal internal tides. The seasonal variability in the HYCOM simulation is also compared with the satellite altimeter data of Zhao (2021).

The seasonal cycle of the semidiurnal SSSH variance is 180° out of phase in the Northern and Southern Hemispheres, which indicates that stratification may be responsible for this seasonal variability. We find that the amplitude of the seasonal cycles is




about 10-15% of the annual-mean values when zonally averaged. The strongest seasonal variability in the semidiurnal SSSH

variance is observed in Georges Bank and the Arabian Sea.

    We compare the seasonal trend in semidiurnal SSSH variance with semidiurnal barotropic to baroclinic energy conversion rate, baroclinic energy flux, KE, APE, and energy. The seasonal trends in the energy terms are quite similar. The conversion rate is dominant in influencing the seasonal variability in the internal tide energetics. However, we observe differences in the seasonal cycles between SSSH variance and the energy terms. Seasonal maxima in energy terms and SSSH do not coincide

in space and time. Moreover, the seasonal cycles in the Northern and Southern Hemispheres are not clearly out of phase as for SSSH. The seasonal cycle of SSSH variance, KE, and, conversion are only similar for Georges Bank and the Arabian Sea, where seasonal variability in internal tides is strong.

    After comparing the seasonal variability in the HYCOM simulation with the satellite altimeter data from Zhao (2021), we find that the seasonal trends in $M_2$ internal tide SSH variance from the satellite altimeter data and the HYCOM simulation

are quite similar. The trend observed in mode 1 $M_2$ KE from the HYCOM simulation is different from both the HYCOM and satellite altimeter mode 1 $M_2$ baroclinic SSH variance for both the Pacific and Atlantic Oceans. Therefore, we conclude that the seasonal variability in KE is different from the internal tide SSH variability.

    Next, we investigate potential mechanisms that may explain the differences in the seasonal variability between semidiurnal SSSH variance and the energy terms. We explore the modulation of SSSH by the seasonal stratification. SSSH is strongly

affected by the density of the surface layers, which varies significantly due to seasonal temperature changes. We compare the seasonal trends in semidiurnal SSSH variance with mode 1 semidiurnal SSH variance, bottom perturbation pressure variance, KE, APE, and buoyancy frequency. Although the seasonal cycles for both mode 1 SSH variance and the undecomposed SSSH variance are similar, they differ from the mode 1 bottom perturbation pressure variance, KE, and APE. The seasonal cycle in the mode 1 SSH variance is mostly due to changes in the mode 1 horizontal velocity eigenfunction at the surface and not due

to changes in the mode 1 perturbation pressure amplitude. The strong stratification in summer causes the horizontal velocity eigenfunction to be surface intensified, which leads to an increase in semidiurnal surface perturbation pressure and SSSH variance.

    According to our analysis, internal tide SSH, KE, and APE are different indicators of the seasonal variability of internal tides in the global ocean. They may represent various dynamic mechanisms of internal tide generation and propagation. However,

we need to conduct further research to understand their relationships and seasonal patterns more accurately.

    Our analysis also suggests that internal tide sea surface height may not be the most accurate indicator of the true seasonal variability of internal tides. Seasonal changes in the surface density stratification can modulate the seasonal variability in sea surface height. Because surface density values and stratification also change on weekly to monthly time scales, it may be possible that the internal tide nonstationarity (Shriver et al., 2014; Zaron, 2017) is overestimated when considering sea surface

height. Nevertheless, sea surface height can still be useful in regions where there is a strong seasonal variability in internal tides, such as the Arabian Sea and Georges Bank.




*Code and data availability.* Hybrid Coordinate Ocean Model (HYCOM) simulations data and code is available at https://zenodo.org/records/10871038 (Kaur and Buijsman, 2024). The satellite altimeter dataset from Zhao (2021) is downloaded from https://figshare.com/articles/dataset/ Seasonal_mode-1_M2_internal_tide_models/14759094.

**Appendix A: Seasonal trends in non-normalized energy terms and steric sea surface height variance**

In this appendix, we show the seasonal trends in the non-normalized semidiurnal barotropic to baroclinic conversion rate, baroclinic energy flux, steric sea surface height (SSSH) variance, baroclinic kinetic energy (KE), available potential energy (APE), and energy for the Pacific and Atlantic Oceans in Figures A1 and A2, respectively. To compute the seasonal trends, we zonally average the conversion rate, flux, SSSH variance, KE, APE, and energy over 10-degree latitude bins for the Atlantic and

the Pacific Oceans for each one-month segment. For all variables, shallow areas are removed (depth < 100 m). The anomalies are computed by removing the annual-mean values. The non-normalized anomalies look similar to the normalized plots in Figures 3 and 4, except at higher latitudes.

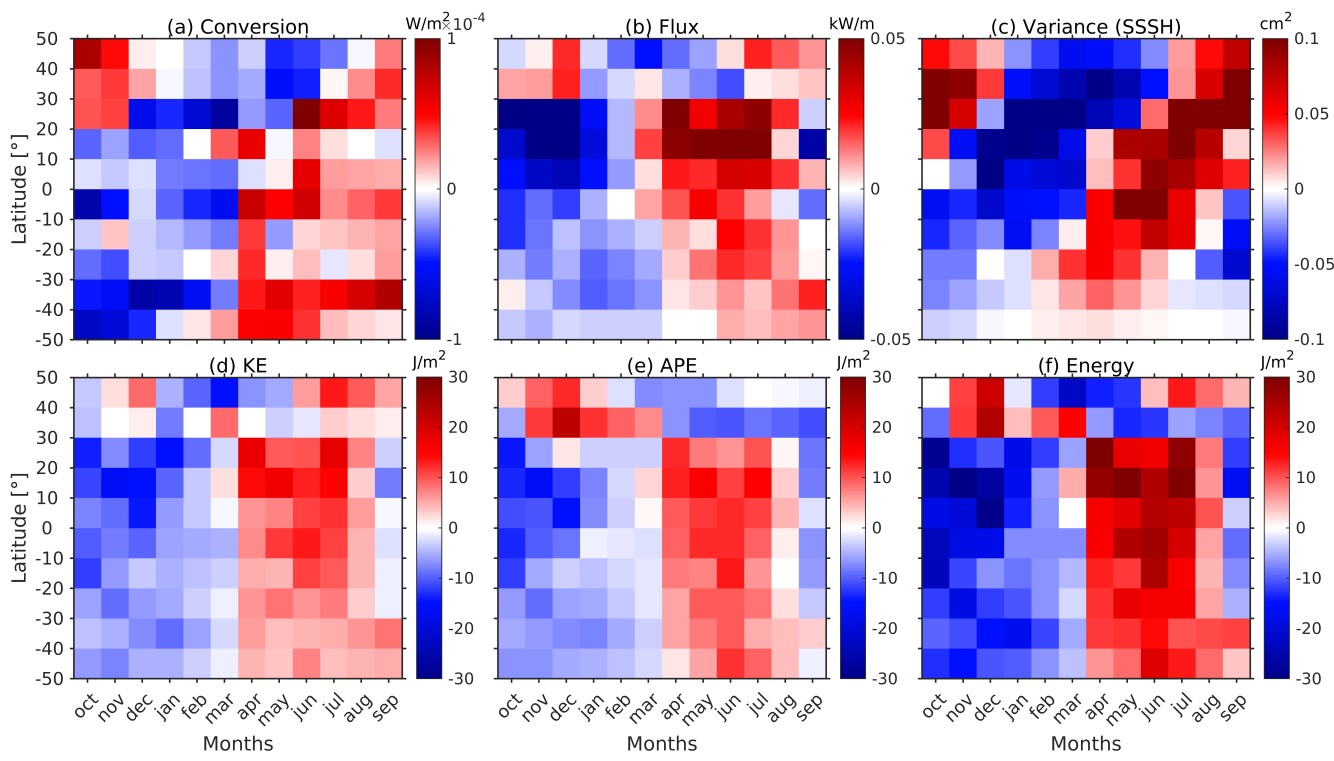

**Figure A1.** Zonally averaged anomaly time series of semidiurnal (a) barotropic to baroclinic conversion rate, (b) baroclinic energy flux, (c) SSSH variance, (d) KE, (e) APE, and (f) energy for the Pacific Ocean.





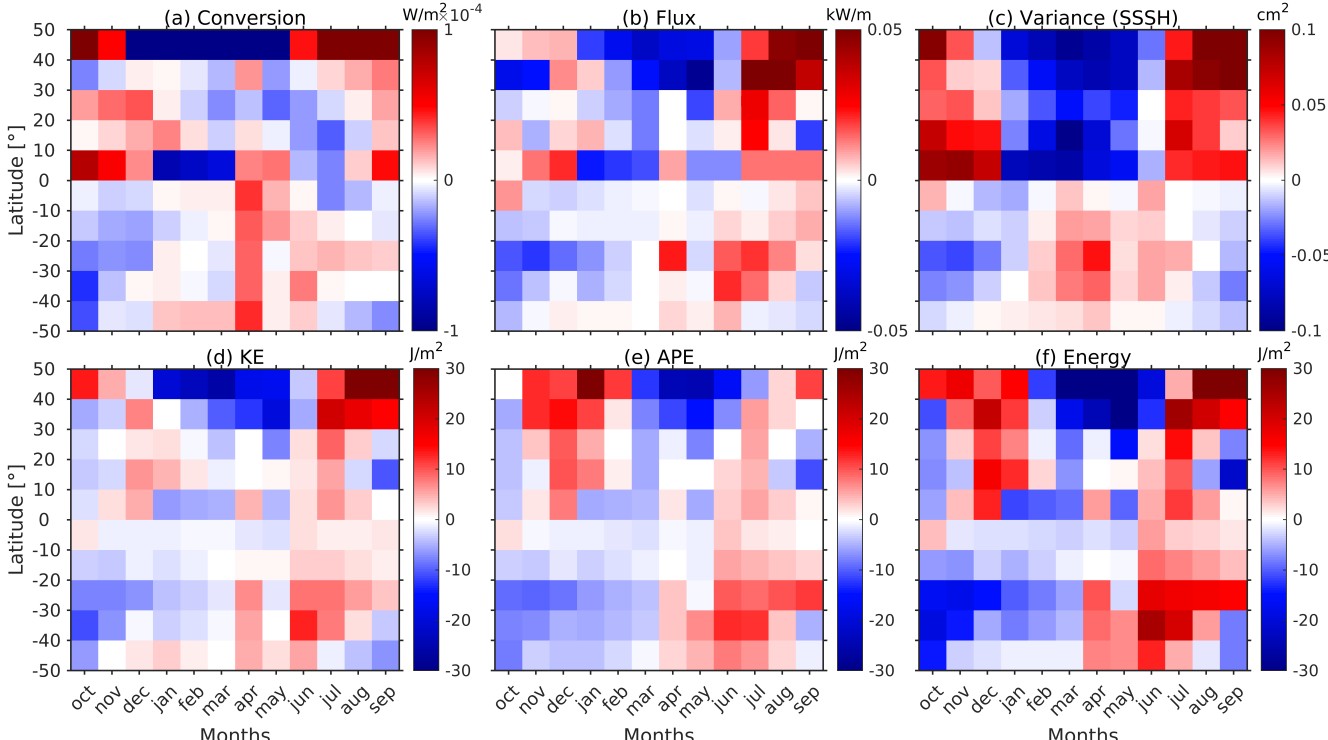

**Figure A2.** The same as Figure A1, but for the Atlantic Ocean.

## Appendix B: Georges Bank and the Arabian Sea

In Section 3.3, it was observed that KE, conversion rate, and SSSH variance exhibit similar seasonal trends in Georges Bank

and the Arabian Sea (Figure 5i, j, o, and p). Furthermore, it was found that seasonal variability is strongest in these two regions (Figure 2a and c). In this section, we conduct a regional analysis of these regions to understand the mechanisms causing seasonal variability in semidiurnal internal tides. Specifically, we investigate the impact of changes in the vertical profile of stratification on the barotropic to baroclinic conversion throughout the year. To do this, we calculate the vertical profile of the conversion rate using the following equation (Kang and Fringer, 2012)

$$C_v(z) = \frac{1}{T} \int_0^T \rho'(z,t) g W(z,t) dt, \tag{B1}$$

where $W(z) = -W(z = -H) \frac{z}{H}$.

Georges Bank and the Gulf of Maine are located in the Northwest Atlantic Ocean. Internal tides are generated on the northeast flank of the Georges Bank and the Northeast Channel in this region (Figure 1b; Chen et al., 2011; Schindelegger et al., 2022). The $M_2$ barotropic tides are strong in this area, but there are only small seasonal changes in barotropic sea surface

height amplitude (Godin, 1995; Katavouta et al., 2016). However, studies have reported seasonal changes in tides related to



stratification in the Gulf of Maine (Chen et al., 2011; Katavouta et al., 2016; Shen et al., 2020; Schindelegger et al., 2022). To understand the mechanisms causing seasonal variability, we compare the conversion rate and buoyancy frequency for the point (294.48°E; 42.09°N) where the conversion rate is at its maximum. We compare the conversion rate calculated by two different methods described in Eq. (6) (method 1) and Eq. (B1) (method 2), and observe that both methods give similar values (Figure

B1a). Additionally, we observe a similar seasonal cycle for the depth-integrated conversion rate and depth-averaged buoyancy frequency at this point. Interestingly, we discover that when stratification is higher near the surface (depths less than 100 m), the conversion rate is also higher for those months throughout the entire water column (Figure B1c and e). We believe this may be due to increased stratification in the surface layer during the summer and fall when solar radiation in the region is at its highest.

In the Arabian Sea region, strong internal tides are generated on the shelf break, which generally propagate offshore as beams (Zhao, 2019; Zaron, 2019; Subeesh et al., 2021; Ma et al., 2021). On the slope, internal tides are stronger in March than in July due to the deepening of the pycnocline during the pre-monsoon period (Subeesh et al., 2021). We compare the vertical profile of the barotropic to baroclinic conversion with buoyancy frequency in Figure B1 for a site (430.64°E; 18.24°N), where the conversion rate is maximum. We get similar results for the conversion calculated using two methods mentioned in Eq. (6)

and Eq. (B1) as shown in Figure B1b. The seasonal trend in depth-integrated conversion rate and depth-averaged buoyancy frequency are not similar. We observe that the conversion rate is large for the months where the magnitude of buoyancy frequency is high at the deeper depths (150-250 m) (Figure B1d and f).

We conclude that seasonal variability in stratification at the generation site is impacting the barotropic to baroclinic conversion for both the Arabian Sea and Georges Bank. However, the surface stratification is responsible for seasonal changes in

Georges Bank, while the vertical profile of buoyancy frequency is the primary factor for the Arabian Sea.

*Author contributions.* HK processed the data, plotted the results, and wrote the first version of the manuscript. MB, ZZ, and JS collected and processed the data. All authors reviewed and edited the paper until its final version.

*Competing interests.* The authors declare that they have no conflict of interest.

*Acknowledgements.* Harpreet Kaur is funded by the National Aeronautics and Space Administration (NASA) grants 80NSSC18K0771 and

80NSSC20K1135, and Office of Naval Research (ONR) USA grant N00014-19-1-2704. Maarten Buijsman is funded by the National Aeronautics and Space Administration (NASA) grants 80NSSC18K0771 and 80NSSC20K1135, and Office of Naval Research (ONR) USA grants N00014-19-1-2704 and N00014-22-1-2576. Jay Shriver was supported by Office of Naval Research (ONR) Grant N0001423WX01413, which is a component of the Global Internal Waves project of the National Oceanographic Partnership Program (https://nopp-giw.ucsd.edu/).



**Figure B1.** (a and b) Time series of depth-integrated conversion and depth-averaged buoyancy frequency for a site in Georges Bank (left column) and the Arabian Sea (right column). Methods 1 and 2 represent the depth-integrated conversion computed using Eq. (6) and Eq. (B1), respectively. (b) Vertical conversion profile and (c) vertical buoyancy frequency profile for a site in Georges Bank (left column) and the Arabian Sea (right column).

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
