# Peer review of "The Seasonal Variability in the Semidiurnal Internal Tide; A Comparison between Sea Surface Height and Energetics"

_EGUsphere, 2024_

## Author Comment (AC2)

**Referee comments 1**

The study of semidiurnal SSH and energetics through HYCOM simulation offers valuable understanding of seasonal variability in the semidiurnal internal tide, , yieding fruitful insights. However, the assertion that seasonal variability in stratification at the generation site affects barotropic to baroclinic conversion in the Arabian Sea and Georges Bank prompts inquiry into the reasons behind the pronounced seasonal variability in these regions compared to other oceanic areas.

>> We thank Referee 1 for their insightful comment. The question raised is intriguing and could serve as a potential topic for future research in this field. However, a detailed analysis falls outside the scope of our current work. We observe that the seasonal variation in barotropic to baroclinic conversion coincides with the seasonal variability in stratification in Georges Bank and the Arabian Sea. The factors affecting stratification in Georges Bank are summer surface heating, surface heat transfer and cold winds during winter, interaction between the Gulf Stream and the southward movement of Labrador Sea water, and advection due to eddies (McLellan, 1957; Gatien, 1976; Brown and Beardsley, 1978; Csanady and Hamilton, 1988; Petrie and Drinkwater, 1993; Katavouta et al., 2016). In the Arabian Sea, the monsoonal winds, which change direction seasonally (Clemens et al., 1991), influence ocean circulation (Shetye et al., 1990, 1991; Beal et al., 2013) and are responsible for changes in pycnocline depth (Rudnick et al., 1997). We now discuss this in Appendix C on lines 458-462 and lines 465-467.

---

## Author Comment (AC3)

**Referee comments 2**

This study investigates the seasonal variability of semidiurnal internal tide steric sea surface height (SSSH) and energetics, comparing trends across hemispheres and geographical hotspots like Georges Bank and the Arabian Sea. Northern and Southern Hemisphere SSSH variance exhibits a phase difference, consistent with altimetry. Seasonal changes in barotropic to baroclinic conversion drive energy variability, while SSSH variance is influenced by seasonal stratification changes. These results are valuable in the internal tide seasonality study. However, there are some questions that need to be discussed. Two main questions are: 1 Why were two simulations run and 2 Is a one-year simulation long enough to do these seasonality analyses?

>> We appreciate the input provided by Referee 2. We first address question 1. We utilize data and data products from two existing Hybrid Coordinate Ocean Model (HYCOM) simulations. While we have a five-year steric sea surface height (SSSH) time series from expt 18.5, we do not have the necessary three-dimensional (3D) fields to calculate the internal tide energy terms. To address this, we have utilized data from a shorter duration simulation, expt 06.1, which provides the required 3D fields from October 2011 to September 2012. As discussed below, the seasonal variability in SSSH of the one-year simulation is very similar to the seasonal variability observed in the five-year simulation. We have modified lines 85-87 to clarify these points and included a third appendix (Appendix A) with a new figure that shows the expt 18.5 has a very similar seasonal variability.

Regarding question 2: yes, a one-year simulation is long enough to perform the seasonal analyses because the seasonal variability in SSSH for the one-year (expt 06.1) and five-year (expt 18.5) simulations are about the same. While Figure 2b for expt 18.5 and Figures 3c and 4c for expt 06.1 show that the semidiurnal SSSH variance in the Northern and Southern hemispheres is out of phase, we acknowledge that the figures and underlying analyses are different. For this reason, we now compute the zonally averaged anomaly time series of SSSH variance for expt 18.5 for all 5 years in a similar way as for expt 06.1. We calculate the semidiurnal SSSH variance for each one-month segment over areas with a seafloor depth greater than 100 m and zonally average the variance over 10-degree latitude bins for the Atlantic and the Pacific Oceans. We have added Appendix A to discuss this. The seasonal variability observed in the five-year time series (Figure A1) closely resembles that of the one-year simulation expt 06.1 (Figures 3c and 4c). Similarly to expt 06.1, the maximum SSSH variance occurs in the Northern Hemisphere in September and October, while the maximum variance in the Southern Hemisphere is observed in March and April.

1. Line 86. Is there any specific reason to select this period to study the seasonality?

    >> We would like to point the Referee to the explanation above. We utilize data from two existing HYCOM simulations. The data from expt 06.1 is available from October 2011 to September 2012; therefore, we use this period.

2. Line 87. If realistic atmosphere forcing was used, is there enough time to spin up the model, to make the model balance?

    >> We thank the Referee for their question. The spin up time of the background circulation and tides for this model simulation (expt 06.1) is 15 years and two months, respectively (Buijsman et al. 2017).

We have added this information on line 96. We also comment on the spin up time of expt 18.5 on line 114.

3. Line 87. Why there are only 5 tidal components are selected? Do more components make more sense?

>> We acknowledge the concerns raised by the Referee. The 5 constituents were chosen by the people who run the simulations because they are easy to separate for a 30-day time series according to the Rayleigh criterion. Additionally, these 5 tidal components are the dominant diurnal and semidiurnal constituents, which contain nearly 95% of all tidal energy (Egbert and Ray, 2003).

4. Line 95. Is this ratio a global average or from a specific region? Would the ratio show a big difference among different regions? So maybe including more tidal components would be better.

>> We thank the Referee for pointing this out. This ratio is global. We have modified the sentence (now line 102) to clarify this. As stated previously, we are utilizing a dataset from an existing simulation, so we are unable to add more constituents. Also, since we use the most dominant semidiurnal constituents, these should be sufficient to capture the semidiurnal signal.

5. Line 99. I didn't get why the author set two simulations (Expt 6.1 and 18.5) with different tidal forcing and under different time coverage. Is there a good reason to explain?

>> Please also see our response to a previous comment. The main reason is that we can only compute the energy terms for the one-year simulation. The one-year and five-year simulations have similar seasonal SSSH variability, which implies that the seasonal variability in the energy terms in the five-year simulation is the same as in the one-year simulation.

6. Line 106. What's the reason for selecting this period?

>> The data from expt 18.5 is available from January 2005 to December 2009; hence we use this period.

7. Line 109. "Steric SSH is computed inline during the HYCOM simulation". Could the author explain more details of this sentence?

>> This statement implies that the steric sea surface height is calculated in real-time as part of the HYCOM simulation. For details on how SSSH is computed we refer the reader to the Appendix of Savage et al (2017). We have modified the text on line 121 to clarify this.

8. Line 112. What does the "both" refer to?

>> We thank the Referee for pointing this out. 'Both' refers to semidiurnal SSSH and true semidiurnal internal tide sea surface height (SSH). We have changed line 125 to make this clear.

9. Function 1. Explain parameter h

>> If the Referee's question is regarding the variable $\eta$ in Eq. 1, then it is the SSSH time series of $M_2$, $S_2$, and $N_2$ internal tide. We have defined it on line 134.

10. Line 128. If my understanding is right, the number 30.42 is calculated by 365/12. However, the author mentioned it is 5 years, so does that mean that using (365*4+366/12) =30.43 makes more sense?

>> Our dataset comprises hourly data points totaling 43824 over 5 years. When we evenly distribute this number by doing 43824/(12*5) months, we get 730.4 data points per month. To fit the annual cycle, we round down to 730 data points per month. This means we have 730 data points per month, which equates to 30.42 days per month. We utilize 730 * 12 * 5 = 43800 data points.

11. Line 143. Could you explain why the approximation is 0?

>> For periodic internal waves in the open ocean, the energy tendency is about zero when averaged over multiple tidal cycles. We have added this information on line 167.

12. Line 164. February has 696 hours, which looks close to the hours in other months. How was the outlier created?

>> We thank the Referee for this question. The values for the energy terms for February were significantly higher in the global ocean. Following Buijsman et al. (2020), we compute the monthly-mean and depth-integrated semidiurnal barotropic to baroclinic energy conversion rate, kinetic energy (KE), available potential energy (APE), and baroclinic energy flux from hourly 3D data of expt 06.1a. Initially, two sets of energy terms are computed for each month. For the first set, we bandpass the monthly time series of the 3D fields between 9 and 15 hours, compute the energy terms every hour, remove the first and last 24 hours to mitigate the effects of ringing, and finally average over time to compute the time-mean energy terms. For the second set, we bandpass the time series of the 3D fields between 9 and 15 hours, we extract the harmonic constants for $M_2$, $S_2$, and $N_2$ constituents, reconstruct the harmonic time series, compute the energy terms every hour, remove the first and last 24 hours to mitigate the effects of ringing, and finally average over time to compute the time-mean energy terms. The order of most steps is similar for both sets, which allows us to compare the total (bandpassed) internal tide energetics of the first set with the phase locked internal tide energetics of the second set (Buijsman et al, 2017). However, at high latitudes mesoscale motions and numerical noise adulterate the energetics of the first set. Hence, we use the second set based on the harmonic time series in this paper. Because February had 29 days, the mean is computed over 27 days, with ~two days of the spring-neap cycle missing. This likely has impacted the energy values for the month of February. We now discuss this in our manuscript on lines 150–161 and lines 190–192.

13. Line 203. Why are the thermobaric instabilities not shown in energy flux? Do these instabilities exist both in Expt6.1 and Expt18.5?

>> These instabilities are present in both expt 06.1 and expt 18.5 (Buijsman et al., 2016; Buijsman et al., 2020). We have added this information on line 238. The thermobaric instability is a non-stationary signal (i.e., not phase locked as the internal tides are), and it is mostly captured when we do a bandpass analysis. In contrast, the least-squares harmonic analysis does not capture these nonstationary signals as much. In addition, the signal projects more on the baroclinic

velocities than on the perturbation pressures. Hence, it is more visible in the KE than in the energy fluxes.

14. Line 219. How was the phase was calculated? What's the meaning of the positive and negative phases?

    >> We thank the Referee for asking this question. This phase is calculated using the least squares fit method (Eq. 3). The phase is $\varphi_a$ in that equation. It is shown in Figure 2b in degrees. If the phase is 90 (-90) degrees, the semidiurnal SSSH variance is maximum in April (October), which implies internal tides are stronger in respective fall months in the Northern and Southern Hemispheres. We have added additional text on lines 252-254 to clarify this.

15. Line 224. Is one year long enough to get reliable seasonal variability?

    >> We would like to point the Referee to our earlier response. Based on our comparison of expt 06.1 with results from expt 18.5 and satellite altimeter data (Figure 6 in the paper), we conclude that a one-year-long simulation is sufficient for seasonal variability analysis.

16. Figure 3. Looks like there is a line at 30 degrees N. At the north of this line, winter and spring shows positive, while at the south of this line is negative. Is there any reason for this?

    >> We concur with the Referee's observation. This may be attributed to the different seasonal variabilities of the wave beams north and south of 30 degrees. The beams to the south have much more energy than the beams to the north. However, after some further investigation, the underlying cause remains unclear.

17. Line 260. What's the percentage described?

    >> This percentage is obtained from the normalized anomaly time series shown in the right column of Figure 5. We have added additional text on line 298 to clarify this.

18. Line 263. Is one year of simulation long enough to get the phase lag conclusion?

    >> We thank the Referee for this question. The phase lag occurs between the SSSH and the energy term time series of the one-year simulation expt 06.1. All energy terms show similar trends that are different from the trends in SSSH (Figures 3 and 4). Hence, we think that for the one-year simulation the phase lag is consistent. We do not have the energy terms for the five-year simulation. Therefore, we cannot directly comment on the phase lag for the five-year simulation. All we can state is that the seasonal variability in semidiurnal SSSH variance is similar between the one and five-year simulations (see the new Appendix A), which is dominated by surface density changes (see previous replies). It is logical to assume that in expt 18.5, the energy and flux variability is also dominated by the conversion variability, which is more sensitive to deeper density changes – as in expt 061. At least we cannot find an argument why that should be different because the seasonal forcing between the simulations is very similar.

19. Line 266-267. Can you do more explain why the conversion rate is the primary factor of the seasonal variability?

>> We consider the conversion rate to be the primary factor influencing seasonal variability because the amount of internal tide energy in the ocean is governed by the internal tide energy input at topography, which is computed with the conversion metric. We have added additional text on line 307 to clarify this.

20. Section 3.4. For the comparison between satellite and simulation. Have you compared the exact energetic value (for example, the energy or energy flux)? Do they under the same magnitude?

>> No, we have not compared the simulated and altimetry-inferred energy terms. The energy and energy fluxes in Zhongxiang Zhao's papers are derived from satellite SSH observations using climatology stratification. In another paper we are preparing, we find that the satellite fluxes are about 20–40% of the model fluxes, because the former contain coherent internal tides only.

21. Line 281. "As" should be "as"?

>> This has been corrected.

22. Figure 6. Did you use the 1992-2017 satellite data to calculate the harmonic parameters? Is the figure 6 (f) can be interpolated from the Figure 4 (d)? Could you mention which experiment you used in the figure caption (for all figures applicable)

>> We utilize satellite altimeter data from Zhao (2021), who used 25 years of multisatellite altimeter data from 1992 to 2017. We have added additional text on lines 221-222 to clarify this.

Figure 4d uses the kinetic energy that is not decomposed into modes (Equation 9). In Figure 6f, we use the mode 1 kinetic energy from Equation 14. We compute the mode 1 kinetic energy for each month and then average it over the months to calculate it for the corresponding seasons. Hence, Figure 6f cannot be interpolated from Figure 4d.

We appreciate the input from the Referee. Experiment numbers are now mentioned in the figure captions.